# Gamma Ray-Induced Polymerization and Cross-Linking for Optimization of PPy/PVP Hydrogel as Biomaterial

**DOI:** 10.3390/polym12010111

**Published:** 2020-01-05

**Authors:** Jin-Oh Jeong, Jong-Seok Park, Young-Ah Kim, Su-Jin Yang, Sung-In Jeong, Jae-Young Lee, Youn-Mook Lim

**Affiliations:** 1Advanced Radiation Technology Institute, Korea Atomic Energy Research Institute, Jeongup-si, Jeollabuk-do 56212, Korea; jojeong86@kaeri.re.kr (J.-O.J.); kya1214@kaeri.re.kr (Y.-A.K.); ysj947812@kaeri.re.kr (S.-J.Y.); sijeong@kaeri.re.kr (S.-I.J.); 2School of Materials Science and Engineering, Gwangju Institute of Science and Technology, Gwangju 61005, Korea; jaeyounglee@gist.ac.kr; 3Department of Polymer Science and Engineering, Chungnam National University, Deajeon 34134, Korea

**Keywords:** hydrogel, polypyrrole, polyvinylpyrrolidone, crosslinking, gamma ray

## Abstract

Conducting polymer (CP)-based hydrogels exhibit the behaviors of bending or contraction/relaxation due to electrical stimulation. They are similar in some ways to biological organs and have advantages regarding manipulation and miniaturization. Thus, these hydrogels have attracted considerable interest for biomedical applications. In this study, we prepared PPy/PVP hydrogel with different concentrations and content through polymerization and cross-linking induced by gamma-ray irradiation at 25 kGy to optimize the mechanical properties of the resulting PPy/PVP hydrogel. Optimization of the PPy/PVP hydrogel was confirmed by characterization using scanning electron microscopy, gel fraction, swelling ratio, and Fourier transform infrared spectroscopy. In addition, we assessed live-cell viability using live/dead assay and CCK-8 assay, and found good cell viability regardless of the concentration and content of Py/pTS. The conductivity of PPy/PVP hydrogel was at least 13 mS/cm. The mechanical properties of PPy/PVP hydrogel are important factors in their application for biomaterials. It was found that 0.15PPy/PVP20 (51.96 ± 6.12 kPa) exhibited better compressive strength than the other samples for use in CP-based hydrogels. Therefore, it was concluded that gamma rays can be used to optimize PPy/PVP hydrogel and that biomedical applications of CP-based hydrogels will be possible.

## 1. Introduction

Conducting polymers (CPs) have excellent electrical properties due to the continuously conjugated structural features of C–C backbones with π–π conjugation. These provide the properties of easy oxidation and reduction by electrochemical changes induced by artificially applying electricity [1,2]. In particular, the excellent electrical properties of the CP-based biomaterials allow it to be used as signal transmission systems for communicating between various cells (e.g., nerve cells, muscle cells, and stem cells) and the materials [3,4,5]. Accordingly, CP-based materials have been applied to materials such as biosensors [6,7,8], neuroprobes [9], tissue engineering scaffolds [3,4], and drug carriers [10,11,12]. This is because the electrical signal can be accurately and incrementally delivered by controlling the degree and duration of the electrical stimulus. CPs are used in a number of representative biomaterials including polypyrrole (PPy), polythiopine, poly(3,4-ethylenedioxythiphene) (PEDOT), and polyaniline [5,13]. As an example, PPy is widely used in biomedical applications, such as biosensors and hydrogels, due to its excellent biocompatibility and environmental stability [3,4,14]. In addition, PPy has the advantage of the range of its electrical conductivity spanning from 10 to 100 S/cm [15]. However, PPy has the disadvantage of being brittle and mechanically unstable due to its conjugated chain structure [16]. Thus, existing CP-based biomaterials have limited use in living bodies because they are too hard and brittle [4,8]. Therefore, it is necessary to develop a biomaterial by which CPs could be applied to the soft tissue of a living body. To overcome such weaknesses, CPs-based hydrogels have been actively developed [17,18,19]. The multifaceted challenge in tissue engineering is to find an ideal hydrogel prepolymer that can mimic the biology of human tissues in terms of structure, function, and performance [20]. One of the major issues in electroactive and conductive tissue engineering is the fabrication of multifunctional hydrogels with native-like biological, electrical, and mechanical properties. Therefore, the incorporation of conductive nanomaterials into hydrogels has gained substantial interest as it not only increases the electrical conductivity, but also improves the elasticity and biological activity relative to the pure hydrogel [21,22].

Hydrogel is a three-dimensional network structure with a polymer chain bonded through covalent and/or secondary bonds, which enables strong hydrogen bonding with water molecules. Hydrogel can contain a large amount of water, and when in contact with water shows substantial swelling. Even so, it does not easily decompose or dissolve in water. In addition, its mechanical strength is easy to adjust because its elasticity and strength vary depending on its structure and its swelling properties mean that its shape in the swollen state is maintained [23,24,25,26,27]. The mechanical properties of hydrogels are tunable both physically and chemically [28,29]. In general, the methods for preparing hydrogel include physical and chemical crosslinking [30,31,32]. Chemical crosslinking methods include the use of chemical reagents and irradiation with electron beams, gamma rays, and ultraviolet light, to activate a polymer via photoreaction involving free-radical formation [32]. In addition, a method for preparing CP-based hydrogels has been studied. It was shown that conductivity could be affected by physically inserting CPs between the hydrogel chains, and then attaching the CPs to the hydrogel using ionic or covalent bonds [17,33].

When cross-linking polyvinyl polymers, the use of gamma-ray irradiation induces free radical formation of both carbon atoms (in the polyvinyl polymer) and hydrogen atoms (in water) that promotes the formation of C–C bonds for cross-linking [34]. The gamma-ray cross-linking method offers many advantages in that reactions are simple and easy to induce in diverse environments (e.g., at low temperature in solid, gas, and liquid states) and without chemical additives (e.g., cross-linking agent, initiators, and catalysts). In addition, gamma ray-induced hydrogels contain no toxic residual substances because chemical additives are not used, and it is possible to control the radiation dose and polymer concentration easily to create hydrogels with a variety of properties [35,36,37]. 

PVP is typically biocompatible, water-soluble, and nonionic polymer that can be applied biomaterials (e.g., hydrogel) due to good solubility and good complex formation [38]. In this study, we fabricated PPy/PVP hydrogels using the gamma-ray cross-linking method. Mixtures of PVP and Py/pTS at different ratios (0.15/0.1, 0.3/0.2, and 0.45/0.3 M) and content (20% and 40%) were exposed to gamma rays. We focused on the optimization of the prepared PPy/PVP hydrogel and confirmed the optimization by characterization using such as scanning electron microscopy (SEM), compressive strength, gel fraction, swelling ratio, Fourier transform infrared spectroscopy (FTIR), and electrical conductivity. In addition, we assessed certain biological properties using live/dead and CCk-8 assays (Figure 1).

## 2. Materials and Methods 

### 2.1. Materials

Polyvinylpyrrolidone (PVP, (C_6_H_9_NO)_n_, molecular weight:360,000), Pyrrole (Py, C_4_H_5_N), para-toluene sulfonate (pTS, C_7_H_8_O_3_S), and aluminum oxide (Al_2_O_3_) were purchased from Sigma-Aldrich (St. Louis, MO, USA). All other reagents and solvents were of analytical grade and used as received.

### 2.2. Preparation of PPy/PVP Hydrogel by Gamma Ray

PVP was dissolved in deionized (DI) water using a magnetic stirrer to a final concentration of 12 wt %. Pyrrole was purified by passing it through an aluminum-oxide column before use. The Py/pTS solution was prepared at different concentration ratios (i.e., 0.15/0.1, 0.3/0.2, and 0.45/0.3 M) in DI water. The preparation at each concentration of the Py/pTS solution was then mixed into the PVP solution (either 20% or 40% Py/pTS). The pure PVP and Py/pTS-containing PVP solution was irradiated using a ^60^Co source (ACEL type C-1882, Korea Atomic Energy Research Institute, Jeongeup, Korea) with a radiation dose of 25 kGy (10 kGy/h) to prepare the PVP and PPy/PVP hydrogel. Table 1 shows the chemical composition of PPy/PVP Hydrogel prepared by gamma-ray irradiation in this study. 

### 2.3. Characterization of the PPy/PVP Hydrogel 

The surface morphologies were investigated using SEM (JSM-6390, JEOL, Tokyo, Japan) with an electron beam of 5 kV. Prior to SEM imaging, the samples were coated with gold for 70 s by sputter-coating to acquire higher-resolution images. To identify the PPy nanoparticles, the elemental analysis of PVP and PPy/PVP hydrogel was investigated using EDS (TM4000, Hitachi, Tokyo, Japan).

The PPy/PVP hydrogel samples were prepared using an 8 mm biopsy punch and then subjected to compressive strength measurements using a texture analyzer (TA-XT2i, Stable Micro Systems Ltd., Godalming, UK) with a load of 1000 N and a working distance of 10 mm from the punch. The compressive strength was measured under 50% and 100% compression based on the decomposition of the hydrogel between the plates of the test machine, at a crosshead speed of 8 mm/min. 

The swelling ratio was measured using dried hydrogel samples prepared using an 8 mm biopsy punch (Integra, Plainsboro Township, NJ, USA). These were immersed in DI water for different time intervals at room temperature, until the swelling reached equilibrium. The swelling ratio of the hydrogel was calculated from the following equation:Swelling ratio (%) = [(*W*_w_ − *W*_d_)/*W*_d_] × 100(1)
where *W*_d_ and *W*_w_ represent the weights of the dry and wet hydrogel, respectively.

Prior to the gel content measurements, the PVP and PPy/PVP hydrogel samples were prepared using an 8-mm biopsy punch (Integra, USA) and then dried until they were completely free of water. The initial weight of the hydrogel was recorded, and then the hydrogel was stirred in DI water for 24 h at room temperature to allow removal of insoluble components. Then, the hydrogel was dried and weighed to calculate the gel content. The gel content in the hydrogel was calculated from the following equation:Gel content (%) = (*W*_f_/*W*_i_) × 100(2)
where *W*_i_ and *W*_f_ represent the initial and final weights of the dried samples, respectively.

The chemical properties of the PVP and PPy/PVP hydrogels were measured using FTIR equipped with an ATR mode (ATR-FTIR, Bruker TEMSOR 37, Bruker AXS, Inc., Karlsruhe, Germany) over the range 650–4000 cm^−1^ at a resolution of 4 cm^−1^ averaged over 128 scans.

### 2.4. In Vitro Cytocompatibility Test

A cytocompatibility test was performed in vitro according to ISO 10993-5, for which the extraction medium was prepared by immersing the PVP and PPy/PVP hydrogels in Dulbecco′s Modified Eagle′s Medium (DMEM) at 37 °C for 24 h to sterilize them, after which solution was extracted by filtering through a 0.22 µm membrane (Sartorius, Ltd., Epsom, UK). NIH3T3 (density of 1 × 10^4^ cells/well) was seeded in a 96-well plate and then incubated in DMEM medium containing 10% fetal bovine serum (FBS) and 1% PS (penicillin and streptomycin) with 5% CO_2_ at 37 °C for 24 h. After being cultured for 24 h, the culture medium was removed and then the extracted solution (diluted by 2× in culture medium) was added to the 96-well plate and incubated with 5% CO_2_ at 37 °C for 24 h. The cytotoxicity was performed by live/dead staining (LIVE/DEAD Viability/Cytotoxicity Kit, Molecular Probes, Inc., Eugene, OR, USA). The culture medium was removed and washed with Dulbecco′s Phosphate-Buffered Saline (DPBS) to stain the calcein-AM and EthD-1 (diluted to 2 μm and 4 μm with DPBS) for incubation at 37 °C for 15 min. After incubation, live/dead images were acquired using fluorescence microscopy (DMI3000B, Leica, Wetzlar, Germany) and images merged using Image J (NIH, Bethesda, Maryland, MD, USA). Cell viability was determined using a CCK-8 assay. The CCK-8 solution was prepared by mixing with DMEM (1:9), and then incubated with 5% CO_2_ at 37 °C for 3 h. The absorbance was recorded at 450 nm using a microplate reader (Powerwave XS, Biotek, Winooski, VT, VT, USA).

### 2.5. Conductivity Measurement

The electrical conductivity of the PVP and PPy/PVP hydrogels was measured using a four-point probe method (Modusystems, Hanam, Korea) for which the linear scan voltammetry was applied from −1 to 1 V. The size of the of the PPy/PVP hydrogel used in the analysis was 1 cm^2^ and the thickness was measured using a Vernier caliper (Mitutoyo, Takatsu-ku, Japan) to determine the resistance from the I-V curve. The electrical conductivity was calculated from the following equation:Conductivity (S/cm) = 1/(Thickness/Resistance)(3)

### 2.6. Statistical Analysis

All tests were performed at least in triplicate and data were presented as the mean ± standard deviation (SD) unless otherwise noted. Statistical significance was judged using one-way analysis of variance (ANOVA) with Tukey′s post-hoc comparison of the means using SigmaPlot software (*p* < 0.05, Systat software, Inc, San Jose, CA, USA).

## 3. Results and Discussion

### 3.1. Prepareation of PPy/PVP Hydrogel by Gamma-Ray

Radiation-induced cross-linking methods have several advantages: (1) the reaction is easily induced by creation of radicals, (2) it is possible to achieve reactions regardless of the environmental conditions, (3) it is not necessary to use chemical agents, and (4) it is possible to control their properties by the radiation dose [39]. We fabricated PPy/PVP hydrogel using gamma rays to manufacture a CP-based hydrogel that passes electrical signals between cells and the CP-based hydrogel. These signals control the mechanical properties of the hydrogel and cytodifferentiation, which are advantages of conductive hydrogel. First, the various experimental concentrations of Py/pTS solution were prepared and then mixed with PVP solution (either 20% or 40%). Then, samples were irradiated using gamma rays to induce cross-linking and polymerization of PVP and PPy, respectively. During the exposure to gamma rays, free radicals are formed and cause PVP and Py to be cross-linked and polymerized concurrently, which is an advantage of radiation-induced techniques. For example, Lee et al. synthesized polypyrrole using gamma ray-induced oxidative polymerization. They found a new way to synthesize polypyrrole that provided advantages such as uniform morphology of the synthesized polypyrrole and high yield of polymerization [40]. In addition, Meltzer et al. identified the mechanism of PVP cross-linking and collagen cross-linking with PVP by gamma-ray irradiation, and confirmed the potential to cause cross-linking by gamma rays without degradation. The mechanism of PVP cross-link by gamma-ray irradiation is that the water molecule formed free radical (e.g., OH·and H·) to break C–H bond of PVP by OH·and/or H·when PVP solution was radiated. The formed radical of PVP was able to cross-link with other PVP radical to form the hydrogel [34]. We referred to the work of Lee and Meltzer and attempted a route capable of simultaneous reaction. 

### 3.2. Characterization of PPy/PVP Hydrogel 

The surface morphology of the PVP and PPy/PVP hydrogels was confirmed by SEM. In general, a hydrogel has a porous structure that might be used to contain various drugs and growth factors. In addition, a hydrogel can maintain a specific drug concentration continuously in the surrounding tissue for a long time (an advantage of hydrogels) [41,42]. As shown in Figure 2, with the PPy/PVP hydrogel, porous structure and PPy particles were found in the hydrogel. On the other hand, it was confirmed that PVP hydrogel has porous structure and smooth surface because there were no PPy particles. It was confirmed that with PPy/PVP hydrogel, when the concentration and content of Py/pTS increases, the amount of PPy particles also increases. It is believed that a large number of PPy particles arise in the PVP hydrogel through the gamma ray-induced simultaneous polymerization of PPy and cross-linking of PVP. As the concentration and content of Py/pTS increased, the SEM image indicated that more PPy particles were generated, which decreased the hydrogel compressive strength due to the brittleness of PPy.

Figure 3a shows optical images of the PPy/PVP hydrogel before and after the press and recovery. In the case of PPy/PVP hydrogel, according to the Py/pTS concentration and content (e.g., 0.15PPy/PVP20, 0.15PPy/PVP40, 0.3PPy/PVP20, and 0.3PPy/PVP40), it was confirmed that the hydrogel returned to its original state after pressing regardless of the concentration and content. On the other hand, 0.45PPy/PVP20 showed a slight breakage of the form of the hydrogel after the press, and in the case of 0.45PPy/PVP40, the hydrogel was completely broken after the press. This indicates that the PPy has very brittle properties [17] that interfere with the PVP cross-linking reaction in ways that do not retain the hydrogel form after the press. In addition, high concentration Py/pTS exposed to gamma rays becomes PPy and forms PVP hydrogel. This shows the simultaneous reactions of polymerization and cross-linking caused by the gamma rays. As shown in Figure 3b, the compressive strength of the PVP was 76.14 ± 1.49 kPa and the compressive strength of 0.15PPy/PVP20 was 51.96 ± 6.12 kPa. These results were confirmed, and indicated a reduction by 25 kPa compared to the PVP hydrogel. In addition, the compressive strength of 0.15PPy/PVP40 was 24.51 ± 1.70 kPa, which confirmed a reduction of approximately two times that of 0.15PPy/PVP20. It is believed that a great deal of Py polymerization occurs as the Py/pTS content increases and compressive strength decreases due to the extreme brittleness of PPy. The compressive strength of 0.45PPy/PVP40 was 13.73 ± 1.34 kPa, confirming very low compressive strength. In addition, it was confirmed that the compressive strength decreases as the concentration of Py/pTS increases. As shown in the optical images, the physical properties of the hydrogels are considered to be very low.

Figure 4a shows the gel fraction of the PVP and PPy/PVP hydrogels with different concentrations and content of Py/pTS. The gel fraction of PVP hydrogel was 84.69 ± 5.01%, and the gel fraction of 0.15PPy/PVP20 and 0.15PPy/PVP40 was 78.87 ± 1.77 and 71.10 ± 3.49%, respectively. Contrary to the compressive strength results, it was confirmed that there was no significant difference in the reduction of gel fraction even if the content of Py/pTS was increased. This suggests that the concentration and content of Py/pTS affects compressive strength but not the gelation rate owing to the brittleness of PPy. In addition, the gel fraction of 0.45PPy/PVP20 and 0.45PPy/PVP40 was 59.46 ± 3.27 and 46.22 ± 9.04%, respectively. When the concentration of Py/pTS was 0.45/0.3 M compared to 0.15/0.1M, the gel fraction was observed to decrease by approximately 20%.

The swelling ratio of the PVP hydrogel was observed to be 21.11 ± 3.73% at 60 min, and the PPy/PVP hydrogel (according to the concentration and content of Py/pTS) was confirmed to have a high swelling ratio at 60 min, as shown in Figure 4c. The swelling ratio of 0.45PPy/PVP40 was 64.32 ± 8.17%, an increase of about 43% compared to the PVP hydrogel. This suggests that the swelling ratio increases as a result of the low gel fraction of 0.45PPy/PVP40. It was also confirmed that the swelling ratio increased with increase of the Py/pTS content from 20% to 40%, and the concentration of Py/pTS (i.e., 0.15/0.1, 0.3/0.2, and 0.45/0.3 M).

To observe their chemical properties, PVP and PPy/PVP hydrogel were investigated using ATR-FTIR, as shown in Figure 4b. The C–H and CH_2_ peaks of the PVP hydrogel were identified at 2945 and 1650 cm^−1^ and the C=O peaks were identified at 1700 cm^−1^ [34]. When the Py/pTS content was 20%, it was confirmed that the intensity of the C–N peaks (at 1415 and 3230 cm^−1^) of PPy/PVP hydrogel increased with the concentration of Py/pTS [14]. It is believed that a large amount of Py is polymerized by the gamma-ray treatment, depending on the increase in the Py/pTS concentration in the PVP hydrogel. In addition, it was confirmed that there is no change in chemical structure except for a change in the peak intensity. It was also confirmed that there was no change in chemical structure due to the gamma rays, even when the PVP hydrogel containing PPy was prepared using gamma-ray irradiation. Thus, gamma ray-induced cross-linking methods could be expected to produce hydrogels having good electrical conductivity.

### 3.3. In Vitro Cytocompatibility Test

To confirm the in vitro cytocompatibility of the PPy/PVP hydrogel variants, each sample was immersed in DMEM without FBS and PS; then the hydrogel was incubated at 37 °C for 24 h to prepare the solution for extraction. The extracted solution was injected into a cultured NIH3T3 plate to incubate cells for 24 h to confirm cell viability for the PPy/PVP hydrogel. This was done using the live/dead and CCK-8 assays. Figure 5a shows confirmation that the PVP hydrogel retained a high number of live cells due to its excellent biocompatibility. The actual cell viability of 95.69 ± 2.46% was confirmed. In addition, it was confirmed that the PPy/PVP hydrogel retained a high number of live cells regardless of the concentration and content of Py/pTS. As shown in Figure 5b, the result of the CCK-8 assay showed that the PPy/PVP hydrogel had high cell viability (>80%) as with the PVP hydrogel. The cell viability of 0.15PPy/PVP20, 0.3PPy/PVP20, and 0.45PPy/PVP20 was 88.53 ± 11.44, 90.10 ± 12.12, and 85.73 ± 8.14%, respectively. When the content of Py/pTS was 40%, the cell viability of 0.15PPy/PVP40, 0.3PPy/PVP40, and 0.45PPy/PVP40 was 86.22 ± 7.16, 84.40 ± 7.16, and 81.14 ± 4.95%, respectively. These results confirm the excellent biocompatibility of PVP and PPy.

### 3.4. Conductivity Measurement

PVP is a biopolymer repesentative of those with excellent biocompatibility, nonionic characteristics, and no conductivity. Even though PVP is used in biomaterial applications in various fields due to its excellent biocompatibility, PVP could not be used for applications requiring conductive biomaterials due to its lack of conductivity. Figure 6 shows that the conductivity of 0.15PPy/PVP20 and 0.15PPy/PVP40 was 13.72 ± 3.77 and 16.91 ± 2.65 mS/cm. When the Py/pTS content was doubled, the conductivity was about 3 mS. This result suggests that the conductivity did not increase much even if the content was increased two times. In addition, the conductivity of 0.3PPy/PVP20, 0.45PPy/PVP20, 0.3PPy/PVP40, and 0.45PPy/PVP40 was 13.97 ± 5.74, 19.36 ± 5.16, 15.93 ± 0.84, and 20.04 ± 1.32 mS/cm, respectively. When the content of Py/pTS was equal (e.g., 20% and 40%), the conductivity increases with increase in the concentration of Py/pTS. However, when a hydrogel is prepared with high concentration and PPy/PVP content, it has limited use in some fields for biomaterial applications due to its weak physical properties. Compared with the PPy/PVP hydrogels prepared under conditions other than 0.15PPy/PVP20, it was confirmed that 0.15 PPy/PVP20 retained both the physical properties and conductivity. Therefore, we believe that 0.15PPy/PVP20 is the most suitable for applications involving conductive biomaterrials.

## 4. Conclusions

The focus of this study was on possible ways to prepare PPy/PVP hydrogel using gamma-ray irradiation. Preparation of PPy/PVP hydrogels using gamma rays caused polymerization and cross-linking to occur simultaneously, thereby creating conductive hydrogels without the need for additional processes. However, although the hydrogel could be prepared simply using gamma rays, the physical properties of the hydrogel were weak due to the brittleness of PPy, as confirmed by its compressive strength. In addition, the properties of PPy/PVP hydrogel were investigated by characterization using SEM, gel fraction, swelling ratio, and FTIR. It was confirmed that the PPy/PVP hydrogels retained high cell viability even when the PPy/PVP hydrogels were prepared using gamma rays. Importantly, 0.15PPy/PVP20 appeared to produce substantially improved conductivity through the PPy and good mechanical properties when optimized by the gamma-ray cross-linking and polymerization reactions. These results provide good evidence, suggesting that PPy/PVP hydrogels have potential as a competitive candidate for prosthetics and smart drug delivery systems.

## Figures and Tables

**Figure 1 polymers-12-00111-f001:**
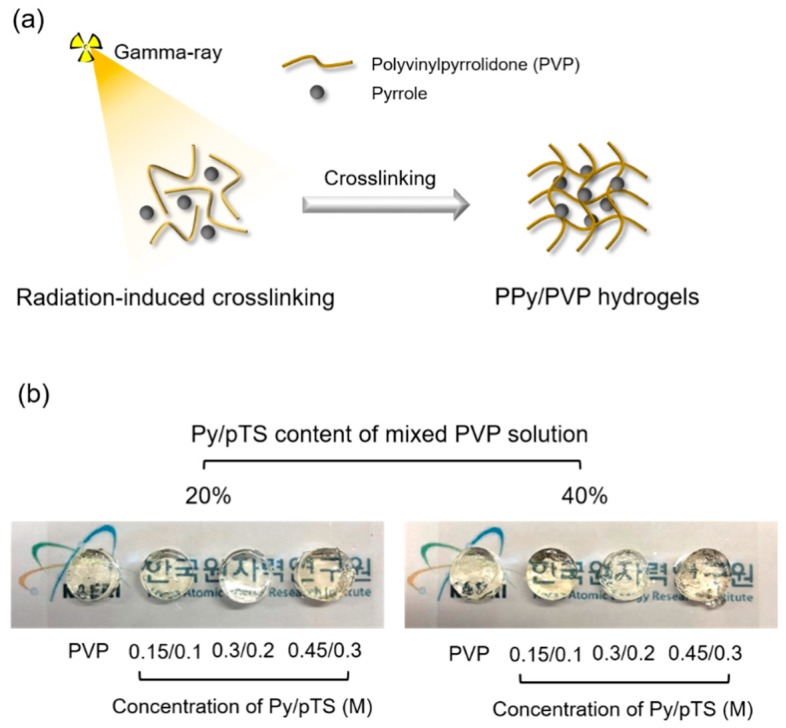
(**a**) Schematic illustration of PPy/PVP hydrogel using gamma ray. (**b**) Optical images of PPy/PVP hydrogel with different concentration and content of Py/pTS.

**Figure 2 polymers-12-00111-f002:**
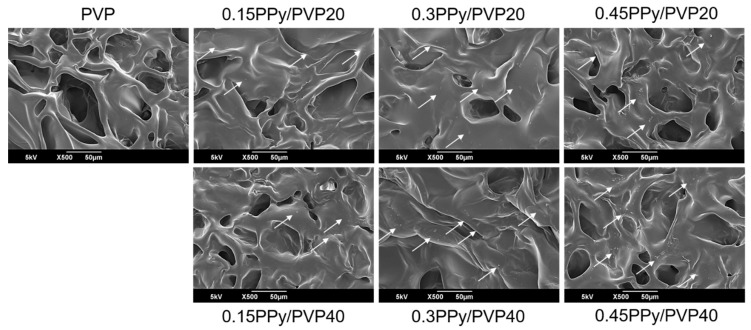
SEM images of PVP and PPy/PVP hydrogels with different concentration and content of Py/pTS.

**Figure 3 polymers-12-00111-f003:**
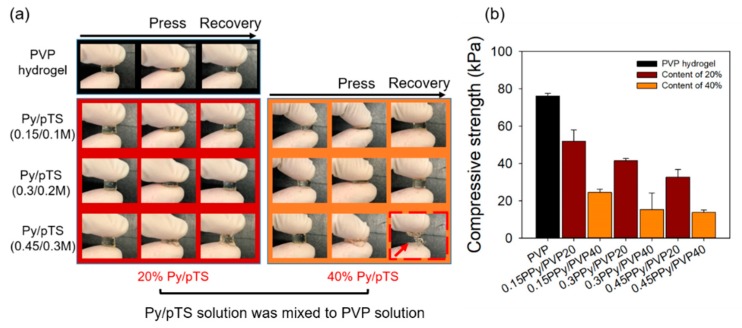
(**a**) Optical images of PVP and PPy/PVP hydrogel after press and recovery. (**b**) Compressive strength of PVP and PPy/PVP hydrogel.

**Figure 4 polymers-12-00111-f004:**
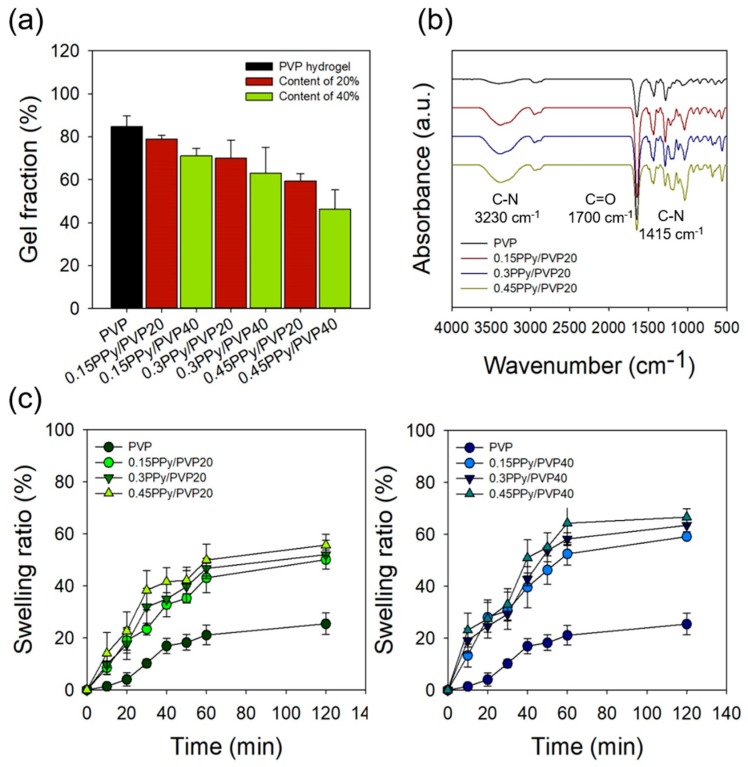
Physical and chemical properties of PVP and PPy/PVP hydrogel (**a**) gel fraction, (**b**) ATR-FTIR of content of 20%, and (**c**) swelling ratio.

**Figure 5 polymers-12-00111-f005:**
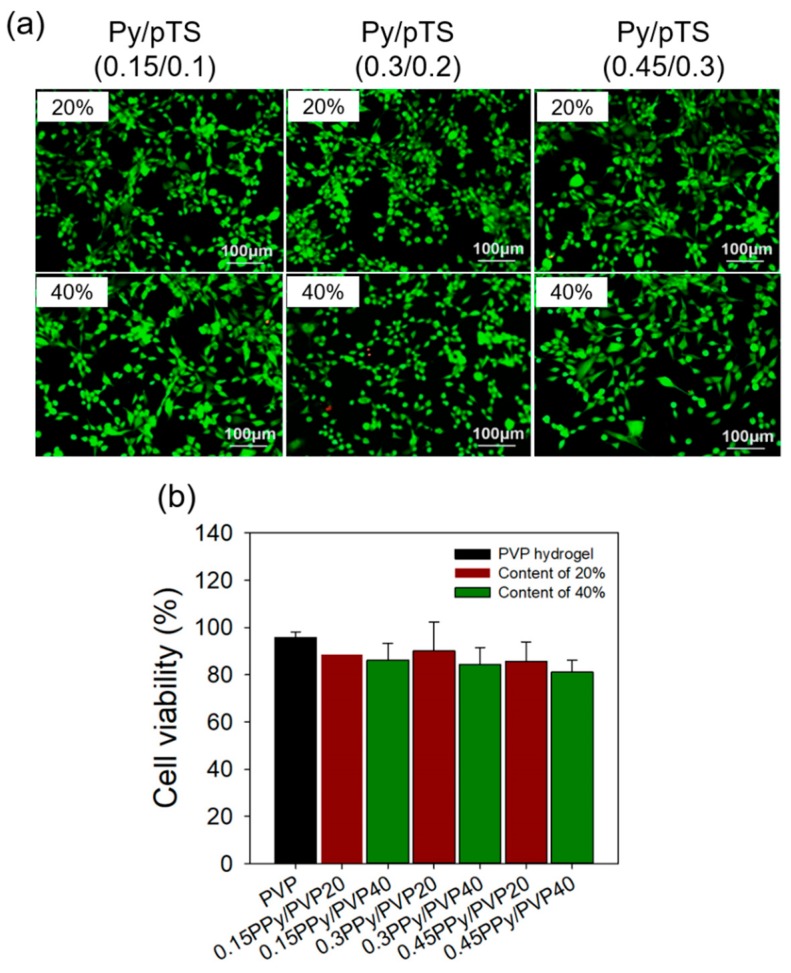
In vitro cytocompatibility of PVP and PPy/PVP hydrogel by live/dead assay using NIH3T3: (**a**) Live/Dead images of PPy/PVP hydrogel with different contentration and content of Py/pTS and (**b**) Cell viability of PVP and PPy/PVP hydrogel using CCK-8 assay.

**Figure 6 polymers-12-00111-f006:**
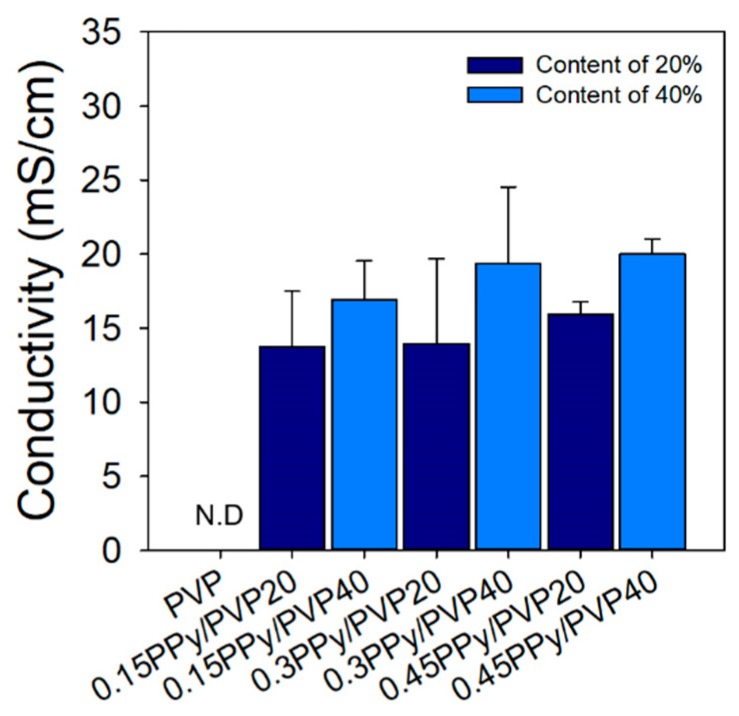
Conductivity of PVP and PPy/PVP hydrogel (N.D = not detect).

**Table 1 polymers-12-00111-t001:** Characteristic of PPy/PVP hydrogels.

Sample	Concentration of Py/pTS (M)	Py/pTS Containing Ratio (%)
PVP	0	0
0.15PPy/PVP20	0.15/0.1	20
0.15PPy/PVP40	0.15/0.1	40
0.3PPy/PVP20	0.3/0.2	20
0.3PPy/PVP40	0.3/0.2	40
0.45PPy/PVP20	0.45/0.3	20
0.45PPy/PVP40	0.45/0.3	40

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
