# Peer review of "Gamma Ray-Induced Polymerization and Cross-Linking for Optimization of PPy/PVP Hydrogel as Biomaterial"

_polymers, 2020, doi:10.3390/polym12010111_

Round 1
Reviewer 1 Report
The manuscript entitled “Gamma-ray Induced Polymerization and Cross-Linking for Optimization of PPy/PVP Hydrogel as Biomaterial” deals with the fabrication PPy/PVP hydrogels by the gamma-ray crosslinking method. The concepts described here are not really novel however they have presented some information that can be useful to the readers. In additional the manuscript is not well written with numerous grammatical mistakes and confusing sentences. Moreover, the methodology must be explain more in detail and results discussion should be analysis clear hypotheses about the data are tested. Therefore the manuscript needs to be revised significantly and there are some queries, which author shall address.
Introduction:
Page 1, line 34-35- the font is different with other texts. Page 1, the authors state that “In the case of CP-based biomaterials, their properties allow them to be 35 used as a signal transmission system for communication using electrical signals between various cells 36 (e.g., nerve cells, muscle cells, and stem cells) and the materials” the statement is not clear. Need to rewrite it more clearly. Page 2, line 43, the properties of PPY need to explain more including advantage and disadvantage of PPY, for example, “the electrical conductivity of PPy can range from 10 to 100 S cm−1 (Blending Electronics with the Human Body: A Pathway toward a Cybernetic Future." Advanced Science 5.10 (2018): 1700931)” and “PPy is brittle and mechanically unstable due to their conjugated chain structure (Bioinspired design of strong, tough, and highly conductive polyol-polypyrrole composites for flexible electronics." ACS applied materials & interfaces 9.7 (2017): 5692-5698.” Page 2, lines 48-59, the introduction about hydrogel is not is not well-written, correct and convincing. For example: “hydrogels strong hydrogen bonding with water molecules” or “In addition, its mechanical strength is easy to adjust because its elasticity and strength vary depending on its structure and its swelling properties mean that its shape in the swollen state is maintained” the authors should mention that the mechanical properties of hydrogels are tunable both physically and chemically (25th anniversary article: Rational design and applications of hydrogels in regenerative medicine. Advanced materials 26.1 (2014): 85-124. And Self‐Healing Hydrogels: The Next Paradigm Shift in Tissue Engineering?." Advanced Science (2019): 1801664.” I recommend reading these articles for find more information about hydrogel: Annabi, Nasim, et al. "25th anniversary article: Rational design and applications of hydrogels in regenerative medicine." Advanced materials 26.1 (2014): 85-124. And Zhang, Yu Shrike, and Ali Khademhosseini. "Advances in engineering hydrogels." Science 356.6337 (2017): eaaf3627. The authors did not any information about PVP in the introduction? I recommend to include this statement in the introduction “The multifaceted challenge in tissue engineering is to find an ideal hydrogel pre-polymer that can mimic the biology of human tissues in terms of structure, function, and performance” and “one of the major issues in electroactive and conductive tissue engineering is the fabrication of multifunctional hydrogels with native‐like biological, electrical, and mechanical properties. Therefore, the incorporation of conductive nanomaterials into hydrogels has gained substantial interest as it not only increases the electrical conductivity, but also improves the elasticity and biological activity relative to the pure hydrogel” References: Pectin Methacrylate (PEMA) and Gelatin-Based Hydrogels for Cell Delivery: Converting Waste Materials into Biomaterials." ACS applied materials & interfaces 11.13 (2019): 12283-12297., Multifunctional nanostructured conductive polymer gels: synthesis, properties, and applications." Accounts of chemical research 50.7 (2017): 1734-1743. And Nanoreinforced Hydrogels for Tissue Engineering: Biomaterials that are Compatible with Load‐Bearing and Electroactive Tissues." Advanced Materials 29.8 (2017): 1603612.Materials and Methods:
The materials and methods are not clear and need to explain more in details and also they mentioned Statistical Analysis but I there is not any Statistical Analysis for cell vitality or other characterization that they have used Statistical Analysis. They should include Statistical Analysis for cell viability and mechanical and physical properties. Moreover, the methodology for obtain conductivity is not clear and need to explain more in details.Results and Discussion:
The SEM images are really unfavorable. The preparation method is really important to obtain goo images and see the pores. The author should use the Cryo‐SEM for imaging and also report the pore sizes. Compressive modulus, breaking point and energy dissipated need to calculate and report. Disintegration rate of the hydrogels need to report.
Author Response
All comments and suggestions are greatly appreciated by authors since these suggestions and comments help us improve this manuscript. We have revised the manuscript carefully according to Reviewer's comment.
Introduction:
Page 1, line 34-35- the font is different with other texts.→ Thanks very much for your valuable opinion. We have revised the font.
Page 1, the authors state that “In the case of CP-based biomaterials, their properties allow them to be 35 used as a signal transmission system for communication using electrical signals between various cells 36 (e.g., nerve cells, muscle cells, and stem cells) and the materials” the statement is not clear. Need to rewrite it more clearly.
→ Thanks very much for your valuable opinion. This sentence is revised as follows "In particular, the excellent electrical properties of the CP-based biomaterials , their properties allow it to be used as a signal transmission systems for communicating between various cells (e.g., nerve cells, muscle cells, and stem cells) and the materials"
Page 2, line 43, the properties of PPY need to explain more including advantage and disadvantage of PPY, for example, “the electrical conductivity of PPy can range from 10 to 100 S cm−1 (Blending Electronics with the Human Body: A Pathway toward a Cybernetic Future." Advanced Science 5.10 (2018): 1700931)” and “PPy is brittle and mechanically unstable due to their conjugated chain structure (Bioinspired design of strong, tough, and highly conductive polyol-polypyrrole composites for flexible electronics." ACS applied materials & interfaces 9.7 (2017): 5692-5698.”
→ Thanks very much for your valuable opinion. We have added the description regarding the properties of polypyrrole with the suggested references in manuscript.
“In addition, PPy has an advantage that the electrical conductivity of PPy can range from 10 to 100 S/cm. However, PPy has the disadvantage of being brittle and mechanically unstable due to its conjugated chain structure.”
Page 2, lines 48-59, the introduction about hydrogel is not is not well-written, correct and convincing. For example: “hydrogels strong hydrogen bonding with water molecules” or “In addition, its mechanical strength is easy to adjust because its elasticity and strength vary depending on its structure and its swelling properties mean that its shape in the swollen state is maintained” the authors should mention that the mechanical properties of hydrogels are tunable both physically and chemically (25th anniversary article: Rational design and applications of hydrogels in regenerative medicine. Advanced materials 26.1 (2014): 85-124. And Self‐Healing Hydrogels: The Next Paradigm Shift in Tissue Engineering?." Advanced Science (2019): 1801664.” I recommend reading these articles for find more information about hydrogel: Annabi, Nasim, et al. "25th anniversary article: Rational design and applications of hydrogels in regenerative medicine." Advanced materials 26.1 (2014): 85-124 . And Zhang, Yu Shrike, and Ali Khademhosseini. "Advances in engineering hydrogels." Science 356.6337 (2017): eaaf3627.
→: Thanks very much for your valuable opinion. We have added the sentence and reference.
The authors did not any information about PVP in the introduction?
: Thanks very much for your valuable opinion. We have added the information of PVP.
“PVP is typically biocompatible, water-soluble, and non-ionic polymer that can be applied biomaterials (e.g., hydrogel) due to good solubility and good complex formation.”
I recommend to include this statement in the introduction “The multifaceted challenge in tissue engineering is to find an ideal hydrogel pre-polymer that can mimic the biology of human tissues in terms of structure, function, and performance” and “one of the major issues in electroactive and conductive tissue engineering is the fabrication of multifunctional hydrogels with native‐like biological, electrical, and mechanical properties. Therefore, the incorporation of conductive nanomaterials into hydrogels has gained substantial interest as it not only increases the electrical conductivity, but also improves the elasticity and biological activity relative to the pure hydrogel” References: Pectin Methacrylate (PEMA) and Gelatin-Based Hydrogels for Cell Delivery: Converting Waste Materials into Biomaterials." ACS applied materials & interfaces 11.13 (2019): 12283-12297., Multifunctional nanostructured conductive polymer gels: synthesis, properties, and applications." Accounts of chemical research 50.7 (2017): 1734-1743. And Nanoreinforced Hydrogels for Tissue Engineering: Biomaterials that are Compatible with Load‐Bearing and Electroactive Tissues." Advanced Materials 29.8 (2017): 1603612.
→: Thanks very much for your valuable opinion. We have added the statement in manuscript
“The multifaceted challenge in tissue engineering is to find an ideal hydrogel pre-polymer that can mimic the biology of human tissues in terms of structure, function, and performance. one of the major issues in electroactive and conductive tissue engineering is the fabrication of multifunctional hydrogels with native‐like biological, electrical, and mechanical properties. Therefore, the incorporation of conductive nanomaterials into hydrogels has gained substantial interest as it not only increases the electrical conductivity, but also improves the elasticity and biological activity relative to the pure hydrogel.”
Materials and Methods:
The materials and methods are not clear and need to explain more in details and also they mentioned Statistical Analysis but I there is not any Statistical Analysis for cell vitality or other characterization that they have used Statistical Analysis. They should include Statistical Analysis for cell viability and mechanical and physical properties. Moreover, the methodology for obtain conductivity is not clear and need to explain more in details.→ Thanks very much for your valuable opinion. The materials and methods were recorded in more detail.
“The electrical conductivity of the PVP and PPy/PVP hydrogels was measured using a four-point probe method (Modysystems, Korea) for which the linear scan voltammetry was applied from −1 to 1 V. The size of the of the PPy/PVP hydrogel used in the analysis was 1 cm2 and the thickness was measured using a Vernier caliper (Mitutoyo, Japan) to determine the resistance from the I-V curve. The electrical conductivity was calculated from the following equation:”
“Conductivity (S/cm) = 1 / (Thickness / Resistance)”
Results and Discussion:
The SEM images are really unfavorable. The preparation method is really important to obtain good images and see the pores. The author should use the Cryo‐SEM for imaging and also report the pore sizes. Compressive modulus, breaking point and energy dissipated need to calculate and report. Disintegration rate of the hydrogels need to report:→ Thanks very much for your valuable opinion.
“In general, a hydrogel has a porous structure that might be used to contain various drugs and growth factors. In addition, a hydrogel can maintain a specific drug concentration continuously in the surrounding tissue for a long time (an advantage of hydrogels).”
The focus of SEM result is to confirm the presence of PPy nanoparticles with increase in concentration of Py/pTS (e.g., 0.15/0.1, 0.3/0.2, and 0.45/0.3) and content of Py solution (e.g., 20 and 40%). In addition, we have showed the elemental analysis of PPy nanoparticles using energy dispersive X-ray spectrometer (EDS) for identification.
Method
Surface morphologies were investigated using SEM (JSM-6390, JEOL, Japan) with an electron beam of 5 kV. Prior to SEM imaging, samples were coated with gold for 70 s by sputter-coating to acquire higher-resolution images. To identify the PPy nanoparticles, the elemental analysis of PVP and PPy/PVP hydrogel was investigated using EDS (TM4000, Hitachi, Japan).
Results
The chemical structure of PVP is composed of (C6H9NO)n and PPy is composed of (C4H2NH)n. Accordingly, as a result of EDS analysis of PVP, carbon, nitrogen, and oxygen were confirmed, and the element amounts were 71.73, 7.34, and 20.91%, respectively. In addition, PPy particles of 0.15PPy/PVP20 were confirmed to carbon (62.93%), nitrogen (19.52%), and oxygen (17.53%) in PVP hydrogel. However, it was confirmed that the amount of elemental nitrogen was increased compared to PVP, and it was considered that the presence of PPy particles existed. As the Py/pTS concentration was increased, the amount of nitrogen was increased. It is believed that as the concentration of Py/pTS increases, a large amount of PPy is formed through polymerization. Through this result, it was confirmed that PPy particles were formed and existed in the PVP hydrogel by simultaneously producing PVP hydrogel using gamma-ray irradiation.
We have added the breaking point of PVP and PPy/PVP.
Method
The PPy/PVP hydrogel samples were prepared using an 8 mm biopsy punch and then subjected to compressive strength measurements using a texture analyzer (TA-XT2i, Stable Micro Systems Ltd., Godalming, UK) with a load of 1000 N and a working distance of 10 mm from the punch. The compressive strength was measured under 50 and 100% compression based on the decomposition of the hydrogel between the plates of the test machine, at a crosshead speed of 8 mm/min.
Results
The gamma-ray induced prepared PVP hydrogel was confirmed the compressive strength of 0.38 kgf and the distances of 15 mm when 100% compression. On the other hand, in the case of PPy/PVP, the compressive strength was decreased compare to PVP and the compressive strength was decreased with increase in concentration of Py/pTS. In addition, it was confirmed that the distance was decreased as the increase in content of Py/pTS. It is believed that was easily braking due to the brittle characteristic of PPy. As Py/pTS concentration and content increase, a large amount of Py was polymerized and thus a large amount of PPy particle was existed in PVP. However, the compressive strength and distance of 0.15PPy/PVP20 were found to be maintained compared to other samples.

Reviewer 2 Report
Should “PVA”at line 89 be “PVP”? Since "PVA" did not appear elsewhere.Figure 2: As a comparison, whether pure PVP hydrogel has also been treated with a radiation dose of 25 kGy? If not,We don't think it has a high persuasive power to prove the particles in the hydrogel is PPy. Please explain it in the text,Or add SEM images of pure PPy particles.
In order to verify the biocompatibility of hydrogels, 3D Cell culture is usually used, or hydrogels-and-cells co-cultured. The methods of“In vitro Cytocompatibility Test”in this article is only an indirect indication that PPy/PVP Hydrogel is less toxic and does not fully demonstrate the cytocompatibility of the material.
If the material is expected to be applied in biomedicine field, it is not adequately by just using NIH3T3. Choosing primary cells for experimentation may be a better option.
Author Response
All comments and suggestions are greatly appreciated by authors since these suggestions and comments help us improve this manuscript. We have revised the manuscript carefully according to Reviewer's comment.
Should “PVA”at line 89 be “PVP”? Since "PVA" did not appear elsewhere.
→ Thanks very much for your valuable opinion. We have revised this sentence
Figure 2: As a comparison, whether pure PVP hydrogel has also been treated with a radiation dose of 25 kGy? If not,We don't think it has a high persuasive power to prove the particles in the hydrogel is PPy. Please explain it in the text,Or add SEM images of pure PPy particles.
→ Thanks very much for your valuable opinion. We were prepared pure PVP hydrogel using gamma-ray irradiation at dose of 25 kGy. We provide supplementary information.
We have added the elemental analysis of PPy nanoparticles using energy dispersive X-ray spectrometer (EDS) for identification.
Method
Surface morphologies were investigated using SEM (JSM-6390, JEOL, Japan) with an electron beam of 5 kV. Prior to SEM imaging, samples were coated with gold for 70 s by sputter-coating to acquire higher-resolution images. To identify the PPy nanoparticles, the elemental analysis of PVP and PPy/PVP hydrogel was investigated using EDS (TM4000, Hitachi, Japan).
Results
The chemical structure of PVP is composed of (C6H9NO)n and PPy is composed of (C4H2NH)n. Accordingly, as a result of EDS analysis of PVP, carbon, nitrogen, and oxygen were confirmed, and the element amounts were 71.73, 7.34, and 20.91%, respectively. In addition, PPy particles of 0.15PPy/PVP20 were confirmed to carbon (62.93%), nitrogen (19.52%), and oxygen (17.53%) in PVP hydrogel. However, it was confirmed that the amount of elemental nitrogen was increased compared to PVP, and it was considered that the presence of PPy particles existed. As the Py/pTS concentration was increased, the amount of nitrogen was increased. It is believed that as the concentration of Py/pTS increases, a large amount of PPy is formed through polymerization. Through this result, it was confirmed that PPy particles were formed and existed in the PVP hydrogel by simultaneously producing PVP hydrogel using gamma-ray irradiation.
In order to verify the biocompatibility of hydrogels, 3D Cell culture is usually used, or hydrogels-and-cells co-cultured. The methods of“In vitro Cytocompatibility Test”in this article is only an indirect indication that PPy/PVP Hydrogel is less toxic and does not fully demonstrate the cytocompatibility of the material. If the material is expected to be applied in biomedicine field, it is not adequately by just using NIH3T3. Choosing primary cells for experimentation may be a better option.
→ Thanks very much for your valuable opinion.
In general, the method of confirming the cell biocompatibility of the hydrogel is a 3D cell culture and cultured using elution solution of the hydrogel. However, in the 3D cell culture suggested by the reviewer, the cell biocompatibility of the hydrogel have difficulty in handling due to the swelling state of the hydrogel by the medium, so it is difficult to confirm the cell biocompatibility in the 3D cell culture. Therefore, in our experiment, the method of confirming cell biocompatibility by elution solution of hydrogel was the most significant, and the experiment was conducted according to the ISO 10993-5 as mentioned in the experimental methods. In addition, the focus of this study was optimization of the preparation of PPy/PVP hydrogels by gamma-ray irradiation and the experiments were carried out using NIH3T3 of fibroblast, which is generally used for cell biocompatibility. Primary cell cultures are currently difficult to experiment. We have to try performed with the experiment later. PVP and PPy are representative biocompatible polymers, and various studies have been conducted for their application as biomaterials. In addition, sterilization proceeds at the same time as the preparation of the hydrogel by gamma-ray irradiation, and there is no toxicity because chemical additives (e.g., crosslinking agent and initiator) are not used.

Reviewer 3 Report
The subject of the manuscript “Gamma-ray Induced Polymerization and Cross-Linking for Optimization of PPy/PVP Hydrogel as Biomaterial” by Jin-Oh Jeong et al. is the production, optimization and characterization of the PPy/PVP hydrogel obtained in the gamma-ray irradiation. The manuscript describes the experimental issues regarding the production of PPy/PVP hydrogel, which can be considered in terms of novelty. The paper is well written and the research work presented in the manuscript falls within the scope of the journal, however the Authors did not avoid few mistakes (which are pointed out below). Moreover the section „Results and Discussion” need to be extended. Authors should pay more attention to properly interpret the results and draw conclusions in an appropriate and comprehensive manner. In the light of above I recommend the manuscript to be published in the Polymers (MDPI) after major revision.
Comments that should be considered by the Authors when revising the manuscript:
As the irradiation method was used for the polymerization process, it could be assumed that the product does not contain impurities (oxidants). Although the Authors could analyze the samples in order to exclude possible pyrrole which has not polymerized or degradation of PVP or PPy due to irradiation. Please provide sufficient data/citations that would clearly indicate that no such changes occurred in this case. Assays done in texture analyzer should be discussed more as one of the main aims is the characterization of mechanical properties. For example instead of providing 'compressive strength' the Authors should include stress-strain plots together with the calculated Young's modulus. The Authors seems to overlook the influence of the temperature on the mechanical properties of hydrogels. It is essential if the hydrogels are to be used as biomaterials (both for sterilization and acting in living organism). Both, chemical and mechanical stability of hydrogels should be assessed as it is essential for further prospect usage. The Authors are asked to provide such data. "Materials and methods", please provide chemical formula, molar mass and melting point/boiling point of all used reagents. Even if they are analytical grade they properties can vary from batch to batch.Author Response
All comments and suggestions are greatly appreciated by authors since these suggestions and comments help us improve this manuscript. We have revised the manuscript carefully according to Reviewer's comment.
As the irradiation method was used for the polymerization process, it could be assumed that the product does not contain impurities (oxidants). Although the Authors could analyze the samples in order to exclude possible pyrrole which has not polymerized or degradation of PVP or PPy due to irradiation. Please provide sufficient data/citations that would clearly indicate that no such changes occurred in this case.
→ Thanks very much for your valuable opinion.
We have added the references for prepared PVP hydrogel and polymerization of pyrrole using gamma-ray irradiation.
On page 5, We have added the mechanism of PVP hydrogel using gamma-ray irradiation to read:
“The mechanism of PVP crosslink by gamma-ray irradiation is that the water molecule formed free radical (e.g., OH· and H·) to break C-H bond of PVP by OH· and/or H· when PVP solution was radiated. The formed radical of PVP was able to crosslink with other PVP radical to form the hydrogel.”
Assays done in texture analyzer should be discussed more as one of the main aims is the characterization of mechanical properties. For example instead of providing 'compressive strength' the Authors should include stress-strain plots together with the calculated Young's modulus.
→ Thanks very much for your valuable opinion. We have added the compressive strength curve of PVP and PPy/PVP.
Method
The PPy/PVP hydrogel samples were prepared using an 8 mm biopsy punch and then subjected to compressive strength measurements using a texture analyzer (TA-XT2i, Stable Micro Systems Ltd., Godalming, UK) with a load of 1000 N and a working distance of 10 mm from the punch. The compressive strength was measured under 50 and 100% compression based on the decomposition of the hydrogel between the plates of the test machine, at a crosshead speed of 8 mm/min.
Results
The gamma-ray induced prepared PVP hydrogel was confirmed the compressive strength of 0.38 kgf and the distances of 15 mm when 100% compression. On the other hand, in the case of PPy/PVP, the compressive strength was decreased compare to PVP and the compressive strength was decreased with increase in concentration of Py/pTS. In addition, it was confirmed that the distance was decreased as the increase in content of Py/pTS. It is believed that was easily braking due to the brittle characteristic of PPy. As Py/pTS concentration and content increase, a large amount of Py was polymerized and thus a large amount of PPy particle was existed in PVP. However, the compressive strength and distance of 0.15PPy/PVP20 were found to be maintained compared to other samples.
The Authors seems to overlook the influence of the temperature on the mechanical properties of hydrogels. It is essential if the hydrogels are to be used as biomaterials (both for sterilization and acting in living organism). Both, chemical and mechanical stability of hydrogels should be assessed as it is essential for further prospect usage.
→ Thanks very much for your valuable opinion. The focus of this study was optimization of the preparation of PPy/PVP hydrogels by gamma-ray irradiation. Confirmation of influence of the temperature on the mechanical properties of hydrogels is currently difficult to experiment. We have to try performed with the experiment later.
The Authors are asked to provide such data. "Materials and methods", please provide chemical formula, molar mass and melting point/boiling point of all used reagents. Even if they are analytical grade they properties can vary from batch to batch.
→ Thanks very much for your valuable opinion. We have added the chemical information in more detail
“Polyvinylpyrrolidone (PVP, (C6H9NO)n, molecular weight:360,000 , meting point:150 oC ), Pyrrole (Py, C4H5N, molecular weight:67.09, boiling point:129 oC), para-toluene sulfonate (pTS, C7H8O3S, molecular weight:172.2 , meting point:38 oC, boiling point:140 oC), and aluminum oxide (Al2O3, molecular weight:101.96 , meting point:2,072 oC, boiling point:2,977 oC) were purchased from Sigma-Aldrich (St. Louis, MO, USA). All other reagents and solvents were of analytical grade and used as received.”

Round 2
Reviewer 1 Report
The authors have addressed comments and questions.
Author Response
All comments and suggestions are greatly appreciated by authors since these suggestions and comments help us improve this manuscript. Thanks very much for your valuable revision opinion.
Reviewer 2 Report
Authors have provided required results.
Author Response

(The authors gave the same response as above.)

Reviewer 3 Report
Thank you for the corrections. In the Reviewer's opinion, the article will contribute to expanding knowledge about polymers obtained by irradiation, and will also be the basis for further scientific discussion.
Author Response

(The authors gave the same response as above.)
